# Fc-Dependent Immunomodulation Induced by Antiviral Therapeutic Antibodies: New Perspectives for Eliciting Protective Immune Responses

**DOI:** 10.3390/antib11030050

**Published:** 2022-07-26

**Authors:** Mireia Pelegrin, Soledad Marsile-Medun, Daouda Abba-Moussa, Manon Souchard, Mar Naranjo-Gomez

**Affiliations:** IRMB, Univ Montpellier, INSERM, CNRS, 34000 Montpellier, France; soledad.marsile-medun@inserm.fr (S.M.-M.); daouda.moustapha-abba-moussa@inserm.fr (D.A.-M.); manon.souchard@inserm.fr (M.S.); maria.naranjo@inserm.fr (M.N.-G.)

**Keywords:** antiviral immunity, antiviral monoclonal antibodies, immunotherapy, FcγR, vaccinal effect, immunomodulation

## Abstract

The multiple mechanisms of action of antiviral monoclonal antibodies (mAbs) have made these molecules a potential therapeutic alternative for treating severe viral infections. In addition to their direct effect on viral propagation, several studies have shown that mAbs are able to enhance the host’s adaptive immune response and generate long-lasting protective immunity. Such immunomodulatory effects occur in an Fc-dependent manner and rely on Fc-FcγR interactions. It is noteworthy that several FcγR-expressing cells have been shown to play a key role in enhancing humoral and cellular immune responses (so-called “vaccinal effects”) in different experimental settings. This review recalls recent findings concerning the vaccinal effects induced by antiviral mAbs, both in several preclinical animal models and in patients treated with mAbs. It summarizes the main cellular and molecular mechanisms involved in these immunomodulatory properties of antiviral mAbs identified in different pathological contexts. It also describes potential therapeutic interventions to enhance host immune responses that could guide the design of improved mAb-based immunotherapies.

## 1. Introduction

Monoclonal antibodies (mAbs) have gained an important place in the therapeutic arsenal against severe human diseases. More than one hundred mAbs have been approved for human use and several hundred are currently being tested in clinical trials [1], most of them to treat patients suffering from a variety of cancers or inflammatory diseases. The development of powerful antiviral mAbs has provided new therapeutic opportunities to treat severe viral infections [2,3,4,5,6], including emerging viral infections. Indeed, the amount of antiviral mAbs is rapidly increasing, with two treatments developed and authorized for the Ebola virus and five for SARS-CoV2 during the years 2020–2021, while other mAbs are currently in development to fight other variants of concern (VOC) [7]. Several lines of evidence show that Fc-dependent mechanisms are crucial for efficient antiviral activity of neutralizing mAbs. Thus, beyond their neutralization capacity, mediated by their Fab fragment upon binding to “vulnerable” viral antigens, the antiviral effect of mAbs is also mediated by the Fc moiety through interaction with the complement system and with Fcγ receptors (FcγRs) expressed by multiple cells of the immune system. This can lead to viral clearance by various Fc-dependent functional responses. The Fc domain allows the binding of complement on antibody-opsonized virions, inducing direct virolysis. FcγR and complement receptors (CR) can recognize opsonized virions, leading to their phagocytosis by cells of the innate immune system. Infected cells can also be eliminated by complement dependent cytotoxicity (CDC), antibody-dependent cellular phagocytosis (ADCP), and antibody-dependent cell-mediated cytotoxicity (ADCC), the latter being mediated by innate immune effector cells expressing the FcγRs. Fc-FcγR interactions can also directly affect viral propagation by other mechanisms, such as antibody dependent cellular viral inhibition (ADCVI) [8]. In addition, FcγR engagement by antiviral mAbs has been shown to have immunomodulatory effects leading to the induction of protective immunity. Indeed, mAbs can form immune complexes (ICs) with different viral determinants (virions or infected cells) that can be recognized by multiple FcγR-expressing cells leading to the induction of stronger antiviral immune responses and the so-called “vaccinal effect” [9,10]. It is worth noting that Fc-FcγR interactions provide a highly versatile system to modulate immune responses (Figure 1). On one hand, there are multiple FcγRs, either activating or inhibitory, which display different affinities for different IgG isotypes. On the other hand, FcγRs are differentially expressed in multiple immune cells, with each of them displaying specific functions. Thus, this diversity allows a myriad of immune functions that can be involved in the control of viral infections, including the induction of protective immunity in an Fc-dependent manner. In this context, the identification of the main FcγRs and the main FcγR-expressing cells, as well as the main Fc-dependent effector functions involved in the induction of protective immunity, is key to maximizing the therapeutic effect of mAbs.

This review summarizes the studies reporting vaccinal effects mediated by antiviral mAbs, initially described in several preclinical animal models and later on observed in mAb-treated patients (in clinical settings). It also focuses on the main cellular and molecular mechanisms involved in these immunomodulatory properties of antiviral mAbs in different pathological contexts, and discusses possible future directions to enhance host immune responses. This topic is now an important field of study, as the possibility of inducing the vaccinal effects by antiviral mAbs is now taken into account by researchers and physicians for the design of improved therapeutic interventions. Importantly, vaccinal effects mediated by anticancer mAbs have also been reported and reviewed elsewhere [11], highlighting that the enhancement of host immune responses by mAbs can occur in different disease settings.

## 2. Multiple FcγR-Expressing Immune Cells Are Involved in the Induction of Vaccinal Effects: Lessons Learned from a Murine Model of Retroviral Infection

The first experimental system to provide mechanistic insight into the generation of protective vaccinal effects by antiviral mAb treatment was the FrCasE retrovirus (a murine leukemia virus, MLV) infection model [9,12]. This model provided the proof of concept that short (five day long) antiviral mAb treatment can induce life-long (more than one year) protective humoral and cellular immunity. This experimental infectious setting has allowed researchers to extensively identify several of the cell types and molecular effectors involved in this process (Figure 2).

The most notable effectors, processes and cells types involved in the induction of vaccinal effects are described below:(i).The vaccinal effects of mAbs strictly depend on Fc-FcγR interactions. In particular, the formation of ICs composed of the administered mAb and infected cells (rather than with virions) enhances the cytotoxic cellular response via the interaction with FcγRs expressed on dendritic cells (DC). These observations also highlighted that the nature of ICs matters to generate protective immunity, as infected cells display immunodominant peptides that are poorly incorporated into virions [13,14],(ii).MAb treatment prevents the development of the regulatory T (Treg) response in an Fc-dependent manner, with specific antibody isotypes involved in such Treg inhibition [15]. Thus, whereas the administration of anti-FrCasE mAbs of the IgG2a isotype prevented the development of Treg responses in infected mice, neither anti-FrCasE mAbs of the IgM isotype nor F(ab’)_2_ antibody fragment administration had the same effect. However, the mechanisms involved in this Fc-dependent inhibition of the Treg response by the therapeutic mAbs were not elucidated.(iii).Neutrophils have a crucial role in the induction of a protective humoral immune response during immunotherapy with neutralizing mAbs [16]. The immunomodulatory potential of neutrophils was evaluated by performing neutrophil depletion experiments. These experiments showed that the absence of neutrophils in infected, mAb-treated mice resulted in a decrease in serum levels of specific anti-FrCasE IgGs as well as a decrease in the frequency of marginal zone B cells and plasma cells in the spleen and bone marrow, respectively. Importantly, neutrophils acquired B cell helper functions upon FcγR-triggering (i.e., secretion of B cell activating factor; BAFF) leading to the induction of a sustained and protective humoral response that was key for the survival of the mice [16].(iv).Neutrophils and monocytes cooperate in the induction of a protective immune response [17]. Notably, upon antibody therapy, neutrophils and inflammatory monocytes display distinct functional activation states and sequentially modulate the antiviral immune response by secreting Th1-type polarizing cytokines and chemokines, which occur in an FcγRIV-dependent manner. Notably, mAb-treatment of infected mice led to a strong upregulation of FcγRIV in neutrophils and inflammatory monocytes, as well as an enhanced functional activation of both cell types (i.e., upregulation of several activation markers and enhanced secretion of cytokines/chemokines). Interestingly, neutrophils showed a higher and a wider induction of chemokines and cytokines release than monocytes at day 8 p.i, while monocytes secreted strong quantities of Th1-polarizing cytokines and chemokines at day 14 p.i., suggesting a potential role for neutrophils as early drivers of the induction of vaccinal effects by mAbs. In addition, FcγRIV-blocking in mAb-treated mice led to decreased secretion of cytokines and chemokines by both myeloid cell-types, as well as reduced mAb-mediated protection.(v).NK cells, in addition to their role in the elimination of infected cells, also have a key immunomodulatory role in the induction of a protective immune response after mAb treatment. This was demonstrated using an NK depletion approach that led to the abrogation of the vaccinal effects induced by mAb therapy (i.e., decreased virus-specific antibody titers and CD8^+^ T cell responses) [18]. The immunomodulatory effects of NK cells are two-fold. Firstly, control of viral propagation by NK cells prevents immune cell exhaustion and the establishment of immunosuppressive mechanisms (i.e., upregulation of molecules involved in immunosuppressive pathways, such as PD-1, PD-L1, and CD39 on dendritic cells and T cells). Secondly, IFNγ-producing NK cells play a role in the enhancement of the B cell responses through the potentiation of the B cell helper properties of neutrophils [18].

Overall, these findings highlight that multiple FcγR-expressing immune cells with specific and complementary functions cooperate to achieve protective immunity upon antibody therapy. This is all the more important to consider as most studies assessing the mechanisms involved in the induction of the vaccinal effect by mAbs mainly point to a role for IC-mediated activation of DC in the enhancement of antiviral T cell responses (reviewed in [10,19]). However, IC-FcγR interactions are not limited to DC, but also concern other FcγR-bearing effector cells of the innate immune system, such as natural killer (NK) cells, neutrophils and monocytes, which have also been shown to participate in the modulation of the antiviral immune response upon mAb treatment.

## 3. Fc-Mediated Immunomodulatory Properties of mAbs Directed against Human Viruses: Evidence from Mouse and NHP Preclinical Models

### 3.1. Hendra and Nipah Henipaviruses Infection

Nipah virus (NiV) and Hendra virus (HeV) are closely related acute and fatal zoonotic viruses within the paramyxovirus genus Henipavirus. Several mAbs have been developed against their F and G glycoproteins that have shown therapeutic efficiency in preclinical models of NiV and HeV infection in ferrets and non-human primates (NHP) [20,21,22,23,24]. Among them, the neutralizing m102.4 mAb, which cross-reacts with both viruses, has been shown to result in the development of antiviral humoral immune responses that correlates with disease recovery upon post-exposure therapy of NiV and HeV infected African green monkeys [20,21] (Table 1). While the mechanisms involved in mounting endogenous humoral responses were not elucidated, these studies suggest an immunomodulatory potential of such anti-NIV and HeV mAbs that might contribute to their therapeutic efficiency.

### 3.2. Acute Respiratory Viral Infections

Several mAbs directed against the influenza virus have been shown to exert potent antiviral activity against diverse influenza strains [31,32,33,34]. The protective effect of such mAbs requires full Fc-effector activity and relies on Fc-FcγR interactions. Recently, with an attempt to identify the cell types and specific FcγRs that contribute to the antiviral activity of anti-influenza mAbs, Bournazos and his colleagues generated several anti-influenza mAbs engineered to display enhanced binding to activating FcγRs. Among the different Fc-engineered mAbs, the GAALIE variant (addition of a three amino acid (G236A, A330L, I332E) modification to the Fc domain), was previously shown to enhance binding to FcγIIa and FcγIIIa receptors, and decrease its affinity for binding to FcγIIb in in vitro experiments [35]. By using a transgenic humanized FcγRs mouse model (recapitulating human FcγR structural and functional diversity) [36], it was shown that selective binding to the activating FcγR (FcγRIIa) resulted in enhanced efficacy to prevent or treat lethal influenza respiratory infections [25]. Importantly, Fc-mediated protection mediated by the GAALIE variant resulted from enhanced DC maturation and the subsequent induction of protective CD8^+^ T- cell responses. No effect on the modulation of humoral responses was observed. The role of other immune cells highly expressing the FcγRIIa, such as neutrophils, was also assessed using a cell-depletion based approach. Depletion of neutrophils had no impact on the antiviral activity of FcγRIIa-enhanced binding mAb variants. However, neutrophil elimination was performed using a single injection of the depleting antibody, which only allows a transient neutrophil depletion followed by a fast restoration of neutrophil counts three days later. Further studies involving sustained neutrophil depletion throughout the presence of the therapeutic antibodies will be required to assess whether or not neutrophils might significantly contribute to the observed FcγRIIa-mediated antiviral protection.

Worthy of note, anti-SARS-CoV-2 mAbs with optimized Fc domains, such as the GAALIE variant, showed superior potency for prevention or treatment of SARS-CoV-2 infection [37]. This was evidenced in animal disease models of COVID-19 that showed improved efficacy of this Fc-variant in both preventing and treating disease-induced weight loss and mortality when compared to a wild-type Fc. However, whether or not the GAALIE variant was associated with the induction of vaccinal effects was not assessed. Nonetheless, Fc-optimized anti-SARS-CoV-2 presenting the GAALIE variant are currently being tested in a clinical trial [37,38] and their therapeutic efficiency versus the parental antibody is being compared.

Fc-mediated modulation of anti-SARS-CoV-2 immune responses has also been shown in mouse and hamster models of SARS-CoV-2 pathogenesis. By using anti-SARS-CoV2 neutralizing mAbs (either in the format of an Fc-variant unable to engage FcγRs or having an intact Fc fragment), Winkler et al. [26] showed that Fc effector functions were not required to protect infected mice when the antibodies were administered as a prophylactic approach. However, intact mAbs reduced SARS-CoV-2 burden and lung disease in mice and hamsters better than loss-of-function Fc variants when mAbs were given after infection. This points to a crucial role for Fc-FcγR interactions in mAb-mediated protection when used as a therapeutic approach. In an attempt to determine which immune cells contributed to the antibody-mediated protection observed in vivo, different FcγR-expressing cells were depleted in further immunotherapy studies. Neither NK nor neutrophil depletion affected the efficiency of intact mAb treatment, suggesting that these cells are dispensable for mAb protection. By contrast, monocyte depletion during mAb therapy was associated with a loss of improvement in lung pathology. In addition, mice treated with the intact mAb showed significant reduction in the numbers of CD45^+^ cells, neutrophils, CD11b^+^DCs, Ly6C^hi^ monocytes in the bronchoalveolar lavage (BAL) at 8 days post-infection when compared to mice treated with the loss-of-function Fc-variant. Decreased counts of myeloid cells were associated with diminished innate immune cells signaling. Thus, Fc engagement of neutralizing antibodies mitigated inflammation and improved clinical outcome by mechanisms that remain to be elucidated. It is worth noting that increased numbers of CD8^+^ T cells and a higher percentage of activated CD8^+^ T cells were observed in animals treated with the intact Fc-mAb (Table 1). The enhanced antibody-dependent CD8^+^ T cell responses observed in mAb-treated animals was suggested to result from enhanced DC activation upon Fc engagement, in agreement with the studies on anti-influenza mAbs mentioned above. However, despite being a plausible explanation, evidence of a role for DC in potentiating the CD8^+^ T cell responses were not reported. Overall, this work points to an Fc-mediated skewing of myeloid cell inflammatory responses in the lungs, as well as a potential role in enhancing antiviral CD8^+^ T cell responses.

These experiments shed light on the Fc-mediated mechanisms involved in the therapeutic effect of human anti-SARS mAbs, in addition to their neutralization capacity against different VOC. Indeed, although passively delivered neutralizing antibodies against SARS-CoV-2 showed therapeutic efficiency against several SARS-CoV2 strains [5], their mechanisms of action in vivo are still ill-understood. A better dissection of the Fc-dependent functional response required to protect patients infected with SARS-CoV-2 (i.e., FcγR, FcγR-expressing cells, required effector functions, etc.) will help to identify key mechanisms that allow increased viral control and/or decreased immunopathology. This might guide the design of improved mAb therapies.

### 3.3. HIV-1 Retrovirus Infection

Several preclinical models of HIV-1 infection in NHPs have enabled the highlighting of the immunomodulatory potential of broadly neutralizing antibodies (bNAbs), in this case anti-HIV-1 antibodies. Thus, enhanced adaptive immunity (i.e., induction of Gag-specific T cell responses) has been reported in different experimental settings involving bNAb-treated, SHIV-infected macaques. Importantly, enhanced frequencies of virus-specific CD4^+^ and CD8^+^ T cells, as well as enhanced functionality (i.e., decreased expression of the exhaustion marker PD-1 on Gag-specific T-lymphocytes), were observed in SHIV-infected macaques upon anti-HIV-1 mAb treatment [27,39] (Table 1). It is worth noting that CD8^+^ T cell depletion in bNAb-treated, SHIV-infected macaques resulted in a viral rebound, suggesting a key role of the bNAb-induced T- cell response in disease protection [28]. As for humoral responses, only a moderate increase in neutralizing antibody titers upon mAb treatment of SHIV-infected macaques was reported, although whether or not it contributed to mAb-mediated protection was not assessed.

Overall, these observations provide evidence that anti-HIV-1 bNAbs can induce vaccinal effects. A key question is now to finely dissect the main cellular and molecular mechanisms involved, including the specific contribution of Fc-FcγR interactions. However, addressing this question is challenging in NHP models because, despite being extremely useful to assess the protective effects of anti-HIV mAbs, their use in the study of immunity is limited due to both technical and cost reasons. In keeping with this, it is important to highlight the differences in activity between the human and macaque Fc receptors that might nuance the interpretation of induced immune responses in bNAb-treated, SHIV-infected macaques [40]. However, several in vitro and mouse in vivo studies shed light on potential mechanisms involved in anti-HIV mAb-mediated immunomodulation. They point to a role for ICs in enhancing anti-HIV-1 adaptive immune responses. Studies involving antibody-opsonized virions showed improved virus-specific CD4^+^ and CD8^+^ T cell responses resulting from enhanced antigen uptake and presentation by dendritic cells (DCs) via FcγRs binding of ICs [41,42,43]. In addition, evidence from immunization approaches using ICs, show a role for the later in shaping antibody responses against HIV-1. Thus, ICs made with gp120 and several anti-HIV antibodies enhanced serum levels of HIV-1-specific antibodies in immunized mice [44,45,46,47]. Furthermore, mice administration of ICs formed with recombinant gp120 proteins and polyclonal antibodies from HIV-infected subjects displaying high neutralization titers induced a high HIV-specific antibody response. The enhanced humoral response was dependent on Fc-fragment properties of the antibodies (notably their glycosylation pattern), as well as on the ICs interaction with complement receptors that led to the acceleration of antigen deposition within B cell follicles [48]. Whether or not such mechanisms also occur in NHP and/ or human patients has not yet been addressed and deserves further investigation.

## 4. Induction of Vaccinal Effects by mAbs in HIV-Infected Patients

The development of powerful anti-HIV-1 bNAbs, able to efficiently control viral propagation and to enhance adaptive immune responses in multiple preclinical models of HIV-1 infection, provided the rationale for studying their capacity to induce vaccinal effects in HIV-1 infected patients. Importantly, the enhancement of both humoral and cellular immune responses by bNAbs has been reported in HIV-1 infected patients [29,30] (Table 1). Schoofs et al. [29] demonstrated that the administration of the therapeutic antibody 3BNC117 to HIV-1 infected patients enhanced infected individuals’ humoral response against the virus during a six-month observation period. Most bNAb-treated, HIV-1-infected patients showed improved antibody responses (with increased breadth and/or potency) to heterologous tier 2 viruses. Furthermore, the elicitation of anti-HIV-1 antibodies occurred in both viremic and aviremic subjects on antiretroviral therapy (ART). By contrast, untreated individuals showed no consistent improvement in their neutralization capacity, neither qualitatively nor quantitatively.

More recently, Niessl et al. [30] showed that anti-HIV-1 antibody therapy is associated with increased virus-specific T cell immunity. In a phase 1b clinical trial, HIV-1-infected individuals on ART were infused with a combination of two bNAbs (3BNC117 and 10–1074) at zero, three and six weeks, followed by temporarily stopping ART (analytical treatment interruption; ATI) two days after the first antibody infusion. Individuals who were infected with HIV-1 and on ART without antibody therapy showed stable or decreasing levels of HIV-1-specific CD8^+^ and CD4^+^ T cell responses over time. In contrast, bNAb-treated patients under ATI showed improved HIV-1 Gag-specific CD8^+^ T cell responses (with significantly increased frequency of antigen-specific CD8^+^ T cells expressing IFN-γ, TNF-α, MIP1-β and/ or CD107A) as well as enhanced CD4^+^ T cell responses (with increased frequency of CD4^+^ T cells expressing IFN-γ, CD40L, TNF-α and/ or IL-2 in response to Gag). Importantly, enhanced CD8^+^ and CD4^+^ T cell responses to Gag were associated with viral suppression for at least fifteen weeks following ATI. However, whether or not these augmented T cell responses contributed to bNAb-mediated viral control was not elucidated.

These results highlight the potential of HIV-1 bNAbs in boosting adaptive immune responses. The elucidation of the mechanism involved and whether such enhanced immune responses might lead to long-term protection in HIV-1 infected patients is now an important issue to address, as it will be key to enhancing the therapeutic efficiency of bNAbs. Several hypotheses have been put forward to explain how antibody therapy boosts the emergence of humoral and cellular immune responses. In particular, it has been hypothesized that the formation of ICs with the therapeutic antibodies and viral determinants, via the engagement of FcγRs on DCs, can lead to enhanced antigen uptake and presentation resulting in the induction of improved antiviral responses. While this hypothesis has been confirmed in in vitro and in vivo mouse studies as detailed above, whether or not this mechanism is involved in bNAbs-treated, in HIV-1-infected patient is still not known.

## 5. Conclusions

Antiviral mAbs have mostly been considered for their neutralization potential. However, several reports in recent years evidenced the possibility of inducing vaccinal effect in different pathological conditions. This has led to a change in the paradigm of the therapeutic effect of antiviral mAb, as the induction of enhanced adaptive immune responses is now sought by recent mAb-based immunotherapies. Several key questions still need to be answered to fully exploit the immunomodulatory potential of therapeutic antibodies. Thus, the fine dissection of the immunological mechanisms that drive the induction of vaccinal effects by antiviral mAbs will guide the design of efficient therapeutic interventions. Different approaches either focused on the improvement of mAb properties or in the use of host-directed therapies might be considered towards this aim. Evidence shows that multiple FcγR-expressing immune cells with specific and complementary functions might be involved in protective immunity during antibody therapy (Figure 2). A better identification of FcγR-expressing cells involved in modulation of immune response will also be key to developing mAbs with improved therapeutic efficiency, notably by generating mAbs carrying Fc variants displaying enhanced binding to specific FcγRs. This is a very intense area of research with significant therapeutic potential [49,50,51]. However, to date, the therapeutic efficacy of Fc-optimized mAbs has primarily been assessed by their ability to mediate enhanced effector functions, while their ability to induce vaccinal effects is still poorly studied. Importantly, as Fc-mediated protective effects might rely on multiple FcγR-expressing cells, preserved cell functions and counts of specific cell types might be required for achieving mAb-induced protective immunity. This opens new prospects for improving antiviral immunotherapies through the use of combined therapies aimed at potentiating or restoring the function of key immune cells. In this regard, it is important to take into consideration that Fc-mediated mechanisms involved in vaccinal effect induction must probably be associated with particular immune mechanisms and inflammatory signatures specific for each pathological situation. Thus, NK and neutrophils were required to both protect retroviral infected mice and enhance antiviral immune responses, but seem to be dispensable in immunotherapies of respiratory viral infections. In addition to this, the use of combinatorial therapies might also rely on counteracting immunosuppressive immune responses and/ or promoting antiviral immune response through the use of different immunostimulatory molecules currently in clinical use. Finally, keeping in mind the specific inflammatory environment and pathological mechanisms associated with different viral infections, the viral and immunological status of infected patients as well as host of intrinsic factors will have to be taken into consideration prior to any therapeutic intervention to achieve efficient viral control while avoiding any potential immunopathology.

## Figures and Tables

**Figure 1 antibodies-11-00050-f001:**
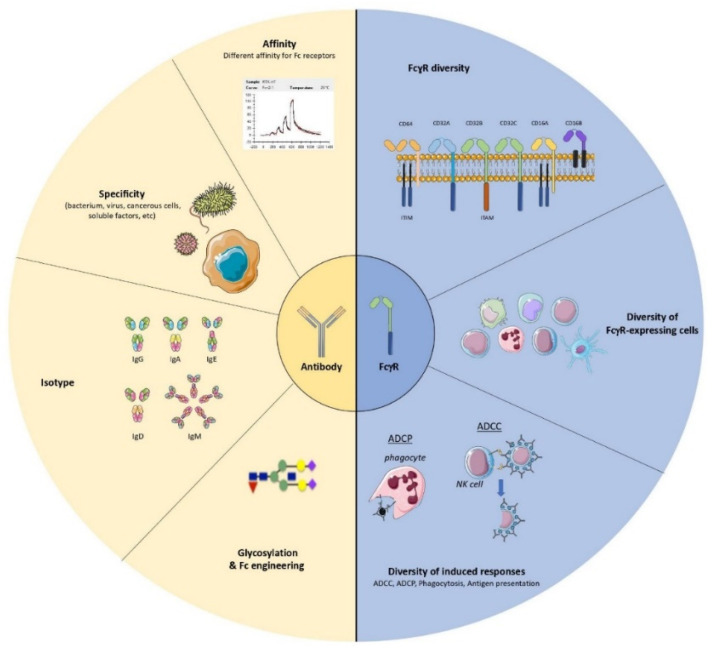
Fc-FcγR interactions constitute a very versatile system to modulate immune responses. The different properties of antibodies (affinity, specificity, isotype, Fc-glyco-engineering, etc.) in addition to the complexity of FcγRs biology (multiple FcγRs, different antibody expression patterns and affinities, multiple FcγR-expressing cells with specific functions, etc.) allow a myriad of immune functions capable of controlling viral spread and modulating immune responses.

**Figure 2 antibodies-11-00050-f002:**
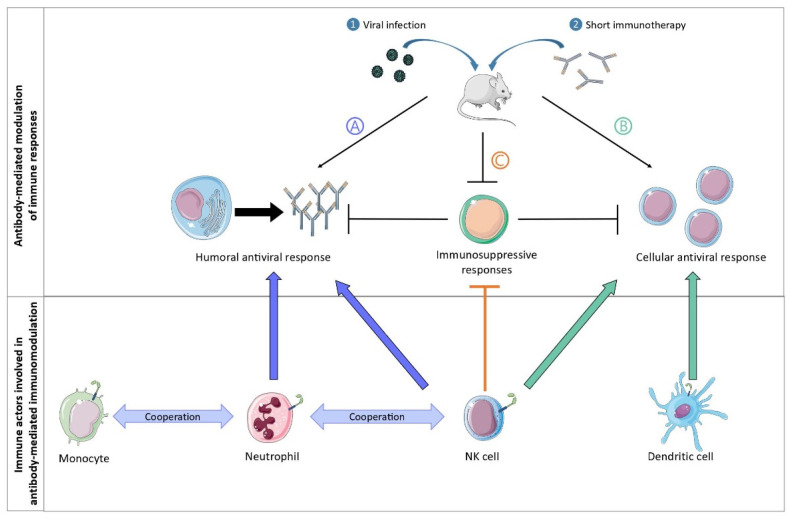
Mechanisms involved in the induction of vaccinal effects. Short treatment of retrovirus-infected mice with a therapeutic monoclonal antibody (mAb) induces a long-term protective response. This is due to (A) the establishment of a humoral antiviral response, (B) the induction of a cellular antiviral response, and (C) the inhibition of immunosuppressive responses (i.e., lack of development of the regulatory T cell response). The mechanisms underlying the induction of protective immunity have been described in this mouse model of a retroviral infection. It has been shown that neutrophils acquire B cell helper functions and are required for the induction of the humoral response (A). Neutrophils also cooperate with monocytes and NK cells to enhance protective immunity. It has also been described that FcγRIV play a key role in the immunomodulatory function of neutrophils and monocytes. Dendritic cells are activated by immune complexes (ICs) formed between the virus and the mAb via their interaction with FcγRs. This results in the enhancement of the antiviral cellular response (B). NK cells are involved in the induction of humoral and cellular responses (A and B), but also, through their ability to control viral spread, they play a role in preventing the development of immunosuppressive immune responses (C) (i.e., the expression of molecules involved in immunosuppressive pathways, such as CD39, PD1, and PD1-L, which are associated with immune cell exhaustion). Thus, several immune cells are involved and may cooperate in establishing a long-term protective immune response.

**Table 1 antibodies-11-00050-t001:** Vaccinal effects reported in preclinical and clinical studies of human viral infections.

Type of Study	Infection	Ab	Animal Model/Patients	Immune Outcome (Observed Vaccinal Effect)	Mechanism	Reference
Preclinical	Henipaviruses	m102.4	African green monkeys	Humoral responses		[20,21]
Preclinical	Influenza virus	3C05 (GAALIE variant)	Transgenic FcγRs humanized mice	CD8^+^ T cell responses	Dendritic cell activation	[25]
Preclinical	SARS-CoV-2	COV2-2050	Mice and hamsters	Increased numbers and more activated CD8^+^ T cells. Decreased inflammation	Potential monocyte involvement in decreasing inflammation	[26]
Preclinical	SHIV-SF162P3	PGT121/3BNC117/b12 mAb cocktail	Rhesus macaques *(Macaca mulatta)*	Increased frequencies and decreased exhaustion of Gag-specific CD8^+^ and CD4^+^ T cells		[27]
Preclinical	SHIV_AD8-EO_	3BNC117 and 10–1074	Rhesus macaques (*Macaca mulatta*)	Polyfunctional CD8^+^ T cells		[28]
Clinical	HIV	3BNC117	Viremic and aviremic subjects on antiretroviral therapy (ART)	Humoral response		[29]
Clinical	HIV	3BNC117 and 10–1074	HIV-1-infected individuals and ART interruption	Virus-specific T cell immunity		[30]

## Data Availability

No new data were created or analyzed in this study. Data sharing is not applicable to this article.

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
