# Peer review of "Fc-Dependent Immunomodulation Induced by Antiviral Therapeutic Antibodies: New Perspectives for Eliciting Protective Immune Responses"

_2073-4468, 2022, doi:10.3390/antib11030050_

Round 1

Reviewer 1 Report

The authors present a short review of Fc-dependent immunomodulation by therapeutic antibodies.  This review highlights some the of added benefits therapeutic antibodies have over more traditional interventions.  While the paper does cover some interesting data, more could be added to make it a better review.  Specifically in section 2 where the authors list immune cells and molecular effectors that are involved in the vaccine effect more should be added.  Although the authors cite references in support of each of the identified  cell types or molecular effectors a description of the experiments that led to theses determinations is lacking.  It would be helpful in the authors supplied more information as was  done for the latter sections of the review. Another example is when the authors describe improved outcomes with the use of the Fc enhancing GAALIE mutations.  While the GAALIE mutations may be the currently favored candidate for Fc enhanced clinical Ab therapy as are  the LS mutations for extended serum half life, there has been a long history of Fc enhancing mutations that have been used (see Saunders, 2019 Frontiers in Immunology). 

Minor points:  There seems to be a formatting problem with the list in section 2.  All gamma symbols are introduced on a new line.  Also on lines 83-83 it should be FcgammaRs and not FcRgammas to be consistent with other references in the review.

Throughout the paper the authors refer to FcgammaRs-interactions.  This should be FcgammaR-interactions without the double plural.  There is only one interaction for each Fc-FcgammaR pair even though there are multiple receptors.

Author Response

We are grateful to the reviewer 1 for her/his constructive critiques. Her/his suggestions have helped us to improve the quality of our manuscript. You will find here below our point-by-point reply to each comment.

To facilitate the reading of the revised manuscript, the major changes in the text are highlighted in yellow.

Comments

“Specifically in section 2 where the authors list immune cells and molecular effectors that are involved in the vaccine effect more should be added”.

- We included in the manuscript additional information in the section (results/methodology). Lines 91-123 of the new version of the manuscript.

While the GAALIE mutations may be the currently favored candidate for Fc enhanced clinical Ab therapy as are  the LS mutations for extended serum half life, there has been a long history of Fc enhancing mutations that have been used (see Saunders, 2019 Frontiers in Immunology). 

- we agree in this comment of the reviewer. This is a very intense area of research with significant therapeutic potential. However, to date, the therapeutic efficacy of Fc-optimized mAbs has primarily been assessed by their ability to mediate enhanced effector functions, while their ability to induce vaccinal effects is still poorly studied.  As this is a very important point, this comment has been included in the conclusion section (lines 312-315) and the reference added. In keeping with this, we detailed the GAALIE mutation, because to our knowledge, it is the only one reported to be associated with the induction of vaccinal effects.

There seems to be a formatting problem with the list in section 2.  All gamma symbols are introduced on a new line. 

Indeed, we have separated each paragraph describing a different mechanism (added line break) for the sake of clarity. Given that additional information has been added, we believe that separating each mechanism will help the reader.

Also on lines 83-83 it should be FcgammaRs and not FcRgammas to be consistent with other references in the review.

- this has been corrected.

Throughout the paper the authors refer to FcgammaRs-interactions.  This should be FcgammaR-interactions without the double plural.  There is only one interaction for each Fc-FcgammaR pair even though there are multiple receptors.

- this has been corrected all over the manuscript

Reviewer 2 Report

This is a timely and well written review concerning recent progress and the future promise of Fc-dependent antibody mediated immunomodulation for protective immune responses, particularly around anti-virals. In particular it focuses on the vaccinal effect and the contribution of the Fc:FcgR interaction therein.

The introduction nicely sets the scope and introduces the Fc effector functions as an adjuvant to the more conventionally considered Fab mediated neutralization.

The mouse model studies followed by human and monkey studies are clearly summarised with the key studies in recent years cited.

My only criticism is that perhaps the field of the vaccinal effect mediated by antibodies in cancer has not been extensively documented and several key studies detailed in the following review might be expanded upon (for strengths and weaknesses) as a useful parallel with the more recently expanded upon viral effects.

https://www.frontiersin.org/articles/10.3389/fimmu.2017.00950/full

Minor

– the statement “More than 100 mAbs have been approved, or are under review, for human use and several hundred are currently being tested in clinical trials [1]” could just be shortened to “More than 100 mAbs have been approved, for human use and several hundred are currently being tested in clinical trials [1]”

-          Line 46 “innate immunity effector cells” perhaps better as “innate immune effector cells”

-          Line 53 “provide a high versatile” better as “a highly versatile”

-          Line 54 - activator or inhibitory” – “activating or inhibitory”

-          Line 80 – unwanted line break to line 81

-          Line 102 – this sentence does not make sense “treatment NK as its depletion a”

The FcgRs in the figure legend to Figure 1 all need s symbol for the g (gamma)

Figure 1 lists ADNP above the phagocyte. should it be ADCP?

Should FcgR be included in the figure 2 ?

-     

Author Response

We are grateful to the reviewer 2 for her/his positive feed-back and constructive critiques. Her/his suggestions have helped us to improve the quality of our manuscript. You will find here below our point-by-point reply to each comment.

To facilitate the reading of the revised manuscript, the major changes in the text are highlighted in yellow.

Comments:

My only criticism is that perhaps the field of the vaccinal effect mediated by antibodies in cancer has not been extensively documented and several key studies detailed in the following review might be expanded upon (for strengths and weaknesses) as a useful parallel with the more recently expanded upon viral effects.

We agree with the reviewer that vaccinal effects mediated by anticancer mAb have also been reported and reviewed in the reference provided (https://www.frontiersin.org/articles/10.3389/fimmu.2017.00950/full). This is important as it highlights that enhancement of host immune responses by mAb can occur in different disease settings. This comment has been included in the manuscript (lines 70-73).  However, as indicated in several sections of the manuscript (abstract, keywords, introduction, conclusion, …) our review was intended to focus on vaccinal effects induced by antiviral antibodies. To avoid any ambiguity, we modified the title of the manuscript precising that we review the immunomodulatory effects of antiviral antibodies.

the statement “More than 100 mAbs have been approved, or are under review, for human use and several hundred are currently being tested in clinical trials [1]” could just be shortened to “More than 100 mAbs have been approved, for human use and several hundred are currently being tested in clinical trials [1]”

- this has been corrected.

-    Line 46 “innate immunity effector cells” perhaps better as “innate immune effector cells”

- this has been corrected.

-          Line 53 “provide a high versatile” better as “a highly versatile”

- this has been corrected.

-          Line 54 - activator or inhibitory” – “activating or inhibitory”

- this has been corrected.

-          Line 80 – unwanted line break to line 81

we have separated each paragraph describing a different mechanism (added line break) for the sake of clarity. Given that additional information has been added according to reviewer 1 comments, we believe that separating each mechanism will help the reader.

-          Line 102 – this sentence does not make sense “treatment NK as its depletion a”

 This sentence has been corrected.

The FcgRs in the figure legend to Figure 1 all need s symbol for the g (gamma)

 -this has been corrected, unless for the “FcgR interactions” according to the comment of reviewer 1, that asked us do not write the double plural “ FcgRs- interactions”

Figure 1 lists ADNP above the phagocyte. should it be ADCP?

-this has been modified.

Should FcgR be included in the figure 2 ?

 -FcgRs have been added on cells. The legend of the figure has also been modified.

Reviewer 3 Report

The review by Pelegrin et al. summarizes the vaccinal effects induced by antiviral mAbs in preclinical animal models and patients mainly due to the Fc-FcgR interactions. As the authors mention, a better understanding of these effects can improve the therapeutic effects of antiviral mAbs. The review is well written with updated and valuable information. Some minor corrections including, for instance:

1.      Lines 86, 89 and 98 should be corrected as the sentences are prematurely broken.

2.      Line 151 should read “mounting endogenous” instead of “in the mounting of the endogenous”  

3.      In line 183 the reference should be “[31].” Instead of “. [31]”

Author Response

We are grateful to the reviewer 3 for her/his positive feed-back and comments. All the minor points mentioned have been corrected.

To facilitate the reading of the revised manuscript, the major changes in the text according to reviewer 1 and 2 comments are highlighted in yellow.

Comments

  1.     Lines 86, 89 and 98 should be corrected as the sentences are prematurely broken.

We have separated each paragraph describing a different mechanism (added line break) for the sake of clarity. Given that additional information has been added according to reviewer 1 comments, we believe that separating each mechanism will help the reader.

  1.     Line 151 should read “mounting endogenous” instead of “in the mounting of the endogenous”  

-this has been corrected.

  1.     In line 183 the reference should be “[31].” Instead of “. [31]”

 -this has been corrected.